# A Counter-Example Based Approach to Probabilistic Conformant Planning

**Anonymous paper**

### Abstract

This paper introduces a counter-example based approach for solving probabilistic conformant planning (PCP) problems. Our algorithm incrementally generates candidate plans and identifies counter-examples until it finds a plan for which the probability of success is above the specified threshold. We prove that the algorithm is sound and complete. We further propose a variation of our algorithm that uses hitting sets to accelerate the generation of candidate plans. Experimental results show that our planner is particularly suited for problems with a high probability threshold.

## 1  Introduction

Probabilistic conformant planning (PCP) (Domshlak and Hoffmann 2006, 2007) is a planning problem in which an agent without observations is tasked to reach the goal with a certain probability guarantee. Because the agent does not have access to observations, the solution to a PCP problem is a sequence of actions. There is uncertainty about the initial state, given by a discrete initial state distribution and uncertainty in the effects of actions; in this paper, we focus on PCP with unknown initial states.

PCP is a generalisation of *conformant planning* (CP) (Smith and Weld 1998) that assumes that the plan should succeed with probability 1. This assumption often means that CP plans are overly conservative in order to account for all unlikely situations: there may be no solutions within the *horizon* (maximum number of actions that we want to allow). Solutions to PCPs for similar domains are often shorter as they are allowed to (selectively) ignore unlikely contingencies. This is similar to the chance-constraints (Birge and Louveaux 2011).

Consider the scenario in Figure 1, where a blind robot wants to reach the destination $(x_2, y_2)$ without prior knowledge of its initial position. Probability distributions to his initial positions are given as follows: $\{(x_1, .2), (x_2, .7), (x_3, .1)\}$ for the x-coordinate, and $\{(y_1, .2), (y_2, .7), (y_3, .1)\}$ for the y-coordinate. Since $x$ and $y$ are independent, we can compute the probability of each initial position by simple multiplication. For example, the probability that the robot starts in $(x_1, y_2)$ is $.2 \times .7 = .14$. At each step, the robot can choose to move left (L), right (R), up (U), or down (D). The grid is surrounded by walls to prevent the robot from moving outside. If the robot hits

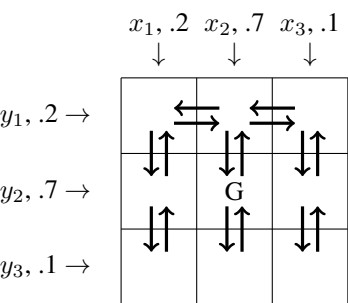

Figure 1: Graphical representation of the GRID problem. The initial location of the robot is unknown with its initial vertical and horizontal locations indicated by the probabilities labeling rows and columns, respectively (e.g., the probability of starting in $(x_1, y_2)$ is $.2 * .7 = .14$). The arrows indicate the possible moves in each state. The goal is to reach the center location with probability .75.

the wall, it will not move but just stand still. We assume that moving left and right is eligible only when the robot is on row $y_1$. What sequence of moves will make the robot arrive at $(x_2, y_2)$ that ensures the combined probability of satisfied initial positions at least being .75? A valid plan for this example is U L R D. This plan works since if the robot initially stands at any location both in the two top-most rows and the two left-most columns, it will eventually reach $(x_2, y_2)$, and the probability of these satisfied initial positions is .81.

### 1.1  Related Work

In principle, PCP problems can be solved by tracking the belief states (probability function over the current state) over all executions up to the length of the planning horizon, and determining whether a goal state was reached with expected probability. In practice, relatively efficient algorithms have been developed that do not exhaustively expand all possible eventualities in the above sense. POND (Bryce, Kambhampati, and Smith 2006) placed in an international planning competition, and implemented a PCP procedure. Its approach involves the application of heuristic search techniques and planning graph heuristics initially developed for non-deterministic planning. POND further uses Sequential

Monte Carlo to generate a set of samples (particles) that represent the distribution over reachable relaxed planning graph layers, and the probability that a particle takes on a particular value is proportional to the probability of that value in the true distribution. The performance of POND is more affected by the number of states, not the complexity of the problem.

Probabilistic-FF (Domshlak and Hoffmann 2006, 2007) is another state-of-the-art planner. This algorithm integrates Conformant-FF's techniques with weighted model counting in (weighted) CNFs to define both the search space and the heuristic function. In particular, Probabilistic-FF extends Conformant-FF belief state formula to construct Bayesian networks, uses weighted model-counting to determine the probability of goals in a belief state, and adapts the heuristic function of Conformant-FF to the probabilistic setting. However, Probabilistic-FF faces challenges in determining the optimal frequency of applying specific actions early in the plan for achieving high confidence in goal attainment. Additionally, the heuristic function of Probabilistic-FF yield bad estimates in some problems.

In 2006, Huang (2006) proposed an innovative algorithm that combines knowledge compilation and search for PCP. The algorithm encodes the PCP problem into a propositional formula, where a subset of propositional variables (chance variables) is assigned probabilities. Subsequently, it compiles these formulas into a Deterministic Decomposable Negation Normal Form (d-DNNF, introduced in Section 4.1), which can be used to compute the satisfied probability of a plan. The algorithm computes the upper bounds on the success probability at each node of the search tree, or for each partial plan generated during search, enabling the pruning of a depth-first search for optimal plans.

Some other PCP are also outstanding. For example, Hyafil and Bacchus (2003) encode the problem by Constraint Satisfaction Problem (CSP) to search a plan; Taig and Brafman (2013) translate PCP problem to classical planning problem and the translation approach is built upon the techniques of Palacios and Geffner (2009); Taig and Brafman (2014) designed another approach in 2014, solving the problem by relevance-based compilation method.

Grastien and Scala (2020) introduced a counter-example based approach CPCES, for solving CP problems. CPCES starts with an empty counter-example set (*sample*) and an empty candidate plan $\pi$. It then iteratively follows two steps: First, it uses a SAT solver to search for a counter-example (initial state) $ce$ that invalidates the current $\pi$, adding $ce$ to the sample; Second, it treanslates the CP problem into a classical planning problem using a multi-interpretation and uses a classical planner to determine a new candidate plan $\pi$. The updated $\pi$ agrees with all the initial states in the sample. CPCES repeats these two steps until either no counter-example can be found, which indicates that the last candidate plan is a valid solution for the CP problem, or no candidate plan can be found, which means there is no solution to the CP problem.

Inspired from CPCES (Grastien and Scala 2020) and Huang's algorithm (Huang 2006), our approach involves encoding the problem using a propositional formula and using a deterministic DNNF to assess the probability of a set of *counter-tags* (defined in Section 3) to a candidate plan. Our algorithm then translates the PCP problem to a classical problem by using *counter-tags* and uses classical planner to find a plan.

This paper is structured as follows: we begin by providing the background of this paper. Then we introduce a counter-example based approach to solving PCP problems. Following that, we detail the process of computing a set of counter-tags and the candidate plan corresponding to these counter-tags. Additionally, we discuss the improvements made to our approach by applying hitting set. Finally, we present the experimental results and conduct an analysis.

## 2  Problem and Background Definitions

Given a propositional formula $\varphi$, we write $vars(\varphi)$ the set of propositional variables in $\varphi$. Every formula is understood as a conjunction $\phi = \bigwedge \phi_i$; it may consist of a single conjunct or even none.

A *PCP problem* is a tuple $\mathcal{P} = \langle F, A, I, G, \tau \rangle$. $F$ is a set of *Boolean facts*. A state $s \subseteq F$ is a subset of *true* facts; this state is also understood as the conjunction $\bigwedge_{f \in s} f \wedge \bigwedge_{f \in F \setminus s} \neg f$ and as the assignment $s[f] = True$ iff $f \in s$. $A$ is a set of *actions*, where each action $a \in A$ is a tuple $\langle name(a), pre(a), coneff(a) \rangle$. Here, $name(a)$ is the unique *action name*, $pre(a)$ is a propositional formula over $F$ called the *precondition*. A precondition is *trivial* if it is *true*. $coneff(a) = \{\langle con_1, eff_1^+, eff_1^- \rangle, \ldots, \langle con_k, eff_k^+, eff_k^- \rangle\}$ is a set of *conditional effects*. In each conditional effect $\langle con, eff^+, eff^- \rangle$, $con$ is a propositional formula called the *condition*; $eff^+$ and $eff^-$ are sets of *positive effects* and *negative effects*, respectively.

The action is applicable in a state $s$ iff $s \models pre(a)$. The positive effects of $a$ in $s$ are $Pos(a, s) = \bigcup_{\langle con, eff^+, eff^- \rangle \in coneff(a), s \models con} eff^+$, and the negative effects $Neg(a, s)$ are computed similarly. The state $s'$ after executing an action $a$ is $s[a] = s \cup Pos(a, s) \setminus Neg(a, s)$. $I : 2^F \to [0, 1]$ is the *initial probability function*; when there is only one initial state $i$, we write $I = \{i\}$ instead of $I = \{i \to 1\} \cup \{s \to 0 | s \neq i\}$. The probability function $I$ is extended to sets of states with $I(S) = \sum_{s \in S} I(s)$. $G$ is a propositional formula over $F$ called the *goal*, which is a condition the algorithm aims to achieve. $\tau$ is a rational number between 0 and 1 and is called the *probability threshold*.

In this article, we assume that each planning problem whose name is decorated is defined as a tuple $\langle F, A, I, G, \tau \rangle$ with the corresponding decoration. For instance, a problem $\mathcal{P}_X^Y$ is implicitly defined as the tuple $\langle F_X^Y, A_X^Y, I_X^Y, G_X^Y, \tau_X^Y \rangle$.

A plan $\pi = name(a_1) \ldots name(a_k)$ is a sequence of action names.[1]

A plan is *valid* for a state $s$ if: *(i)* each action $a_j$ is applicable in state $s[a_1]...[a_{j-1}]$, and *(ii)* it leads to a goal state:

---

[1]Traditionally, a plan is instead defined as a sequence of actions; using action names makes it possible to compare plans for problems that feature different actions—for instance because they are defined over a different set of facts—but the same action names.

$s[\pi] \models G$. We write $[\![\pi]\!]$ the set of states in which $\pi$ is valid. A plan $\pi$ is *valid* for the problem $\mathcal{P}$, written $\pi \in \Pi(\mathcal{P})$, if the probability of it being valid in the initial state is at least $\tau$: $I([\![\pi]\!]) \geq \tau$.

The problem in Figure 1 includes six facts $F = \{x_1, x_2, x_3, y_1, y_2, y_3\}$ and 4 actions $A = \langle \text{L}, \text{R}, \text{U}, \text{D} \rangle$ (left, right, up, down). Actions L and R have precondition $y_1$ – I.e., the agent can only act to move horizontally when located in the top row. There are nine initial states. The goal is $x_2 \wedge y_2$ and $\tau = .75$.

The initial probability function is

$$I = \left\{ \begin{array}{lll} \{x_1, y_1\} \rightarrow .04 & \{x_1, y_2\} \rightarrow .14 & \{x_1, y_3\} \rightarrow .02 \\ \{x_2, y_1\} \rightarrow .14 & \{x_2, y_2\} \rightarrow .49 & \{x_2, y_3\} \rightarrow .07 \\ \{x_3, y_1\} \rightarrow .02 & \{x_3, y_2\} \rightarrow .07 & \{x_3, y_3\} \rightarrow .01 \end{array} \right.$$

and $I(s) = 0$ for any other state.

A PCP problem with $\tau = 1$ corresponds to a CP problem. A PCP problem with only one initial state $i$ is equivalent to a *classical planning problem*.[2]

We now present the concepts of tag and context (Palacios and Geffner 2009). We define a *subgoal* to be a conjunct of the goal formula, or some action precondition. A fact $f_1$ *depends* on another fact $f_2$ if there is a conditional effect $\langle con, eff^+, eff^- \rangle$ that satisfies $f_2 \in vars(con)$ and $f_1 \in eff^+ \cup eff^-$. A *context* $c(\phi)$ of a subgoal $\phi$ is the union of facts $vars(\phi)$ with all other facts they depend on, transitively. In other words, the context of a subgoal includes all the facts whose truth in the initial state can influence the validity of the subgoal in any state during the execution. Given a context $c$, the *tag* of initial state $i$ is the intersection of $i$ with $c$: $t_c(i) = i \cap c$; we generally add the context to the tag and write the tag as a pair $ct = \langle c, t \rangle$. The set of tags of $\mathcal{P}$ is $Tags(\mathcal{P})$.

Our running example of Figure 1 features three subgoals: $x_2$ and $y_2$ (i.e., each from the goal), and $y_1$ (precondition of actions L and R). Consider ssubgoal $\phi = y_1$. Here, $y_1$ depends on $y_2$ from conditional effect $\langle y_2, \{y_1\}, \emptyset \rangle$ of U, and $y_2$ depends on $y_3$. Hence, the context of $\phi$ is $\{y_1, y_2, y_3\}$, the facts characterising the agent's vertical position. This means that, to decide if action L is applicable after applying the known sequence of actions $a_1, \ldots, a_k$ from the unknown initial state $i$, knowing the vertical location of the robot in $i$ suffices.

## 3 A Counter-Example Based Approach

In this paper, we develop a counter-example based approach similar to CPCES (Grastien and Scala 2020). In our running example, from sample $\{(x_2, y_1), (x_2, y_2)\}$, CPCES might propose candidate plan $\pi = \text{UD}$, a sequence of action valid for both states. This plan is not conformant because it is not valid in state $(x_1, y_3)$. If that state is added to the sample, the candidate plan at next iteration might be U U L R D.

One could be tempted to use a similar approach for PCP problems with the difference that a counter-example is generated only if the probability of success $I([\![\pi]\!])$ for current

---

[2]Note, in this case a sequential execution will achieve the goal with probability 1 or 0.

| # it | Counter-tags | CTS | Proba | Cand. plan. |
|---|---|---|---|---|
| 0 | - | - | - | $\varepsilon$ |
| 1 | $x_1, x_3, y_1, y_3$ | $\{x_1, x_3\}$ | .3 | L |
| 2 | $x_1, x_2, y_1, y_2, y_3$ | $\{x_2\}$ | .7 | L R |
| 3 | $x_3, y_1, y_2, y_3$ | $\{y_2\}$ | .7 | U R L D |
| 4 | $x_1, y_3$ | $\{x_1, y_3\}$ | .28 | U L R D |
| 5 | $x_3, y_3$ | None | N/A | - |

Table 1: Example execution of OURALGO for the running example with $\tau = .75$.

candidate plan $\pi$ is below the threshold. However, this procedure is not complete. Indeed, if there is a state from which the goal is unreachable, then adding this state to the sample will corrupt it. This is because, if this procedure is to iteratively find plans with higher probability of success, then each call to the classical planner must pose a soluble classical problem. By adding states from which the goal is not reachable to the starting conditions, the posed classical problem is not soluble.

In our running example, if we assume that there is a wall from the bottom row to the middle row—i.e., the robot cannot move from $y_3$ to $y_2$— then adding state $(x_1, y_3)$ would mean that no more candidate plan can be generated although the problem is solvable for $\tau \leq .9$. More generally, the sample could be corrupted by a combination of states.

Instead, we build on top of the tag/context theory (Palacios and Geffner 2009). We say that a tag is a 'counter-tag' (introduced in Section 4) to a candidate plan if the plan always fails to achieve one of its subgoals associated with the context when the initial state satisfies the tag. Counter-tags implicitly represent large groups of initial states (potentially exponentially large ones) in which the plan is invalid. Our algorithm (OURALGO) computes a 'counter-tag set' (CTS) that covers the initial belief with probability at least $(1 - \tau)$. Then, when computing candidate plans in later iterations, OURALGO makes sure that the plan is valid for (or *consistent with*) at least one of these tags.

Table 1 illustrates the execution of OURALGO for our running example with $\tau = .75$. Notice that since all tags are singletons, we represent them with facts instead of sets: when we say 'tag $x_1$', we actually mean 'tag $\{x_1\}$ over context $\{x_1, x_2, x_3\}$'.

0. The first candidate plan is the empty plan $\pi_0 = \varepsilon$.

1. The counter-tags to $\pi_0$ are $x_1$, $x_3$, $y_1$, and $y_3$: e.g., executing $\pi_0$ for an agent starting in $x_1$ will fail to achieve the goal. A counter-tag set is $\{x_1, x_3\}$, as these two tags add up to probability $0.3 > 1 - 0.75$. From that point on, the candidate plans should be such that either $x_1$ or $x_3$ will not be a counter-tag. OURALGO chooses $\pi_1 = \text{L}$ which is consistent with $x_3$.

2. The table lists the counter-tags to $\pi_1$; note that $y_2$ is a counter-tag as action L cannot be applied when the agent is in $y_2$. A counter-tag set is $\{x_2\}$ with probability mass .7. Since this CTS contains only one tag, future plans will all make sure that $x_2$ is no longer a counter-tag. In this round, $\pi_2 = \text{LR}$ is chosen as candidate plan, which

---

**Algorithm 1:** OURALGO

1: **Input:** PCP problem $\mathcal{P}$
2: **Output:** a plan $\pi$ for $\mathcal{P}$, or UNSAT
3: $B := \emptyset$               $\triangleright$ a set of counter-tags
4: $\pi := \varepsilon$                $\triangleright$ candidate plan
5: **loop**
6:    $CTS :=$ compute_CTS$(\mathcal{P}, \pi)$     $\triangleright$ explained in Section 4
7:    **if** $CTS = \bot$ **then return** $\pi$
8:    $B := B \cup \{CTS\}$
9:    $\pi :=$ compute_candidate_plan$(\mathcal{P}, B)$   $\triangleright$ explained in Section 5
10:    **if** $\pi = \bot$ **then return** no plan

---

is consistent with both $x_1$ and $x_2$. Notice that OURALGO chooses a plan consistent with $x_1$, while it chose $x_3$ in the first iteration: unlike CPCES, OURALGO does not commit to a state/tag.

3. OURALGO keeps running and finds increasingly sophisticated candidate plans.

4. Consider the fourth counter-tag set, $\{x_1, y_3\}$. Note that while $x_1$ represents probability $p_1 = .2$ and $y_3$ probability $p_2 = .1$, their combined probability is not $p_1 + p_2 = .3$ but .28. OURALGO computes plan $\pi_4 = $ U L R D.

5. Finally, $\{x_3, y_3\}$ are counter-tags to $\pi_4$, but their combined probability is 0.19, which is less than 0.25. As no CTS exists, the last candidate plan $\pi_4$ must be a valid plan. Indeed, $\pi_4$ has probability of success of $.81 \geq \tau$.

OURALGO is summarised in Algorithm 1. $B$ collects the counter-tag sets computed so far, and candidate plan $\pi$ starts with $\varepsilon$. OURALGO computes a new counter-tag set; if no such set exists, the current candidate plan is a solution. Otherwise, the set is added to $B$. Next, OURALGO computes a candidate plan that is consistent with at least one tag of each counter-tag set from $B$; if no such plan exists, then there is no solution to the planning problem. Otherwise, a new iteration starts.

## 4 Computing Counter-Tags

We define counter-tags and explain how to verify if a tag is a counter-tag. As described in Section 3, counter-tags are useful in that *(i)* they allow one to talk about many states in which a plan is invalid without enumerating these states, and *(ii)* they help find the next candidate plan.

The definition of counter-tags is based on the notion of a projected planning problem. Projection restricts the planning problem to a context, and sets the initial state to be a tag. We define the projection bottom up, from simple objects to complex ones.

**Definition 1** *Let $ct = \langle c, t \rangle$ be a tag.*

- *The projection of a conjunction $\varphi = \bigwedge_{j \in \{1,\dots,k\}} \varphi_j$ over $ct$ is $Proj(\varphi, ct) = \bigwedge_{j \in \{1,\dots,k\}.vars(\varphi_j) \subseteq c} \varphi_j$.*
- *The projection of a set of conditional effects coneff over $ct$ is the subset of conditional effects $Proj(coneff, ct) = \{\langle con, c \cap eff^+, c \cap eff^- \rangle \in coneff \mid eff^+ \cup eff^- \subseteq c\}$.*

- *The projection of action $a$ over $ct$ is $Proj(a, ct) = \langle name(a), Proj(pre(a), ct), Proj(coneff(a), ct) \rangle$.*
- *The projection of $\mathcal{P} = \langle F, A, I, G, \tau \rangle$ over $ct$ is the classical planning problem $Proj(\mathcal{P}, ct) = \mathcal{P}_{ct}$ defined by $F_{ct} = c$, $A_{ct} = \{Proj(a, ct) \mid a \in A\}$, $I_{ct} = \{t\}$, and $G_{ct} = Proj(G, ct)$.*

A counter-tag to a plan is a tag such that the plan is invalid in the projected problem.

**Definition 2** *A counter-tag to $\pi$ is a tag $ct \in Tags(\mathcal{P})$ such that $\pi \notin \Pi(Proj(\mathcal{P}, ct))$. The set of all counter-tags to $\pi$ is $CTags(\pi)$.*

To verify if a tag is a counter-tag to a plan, one can simply simulate the execution of the plan and check that each precondition and the goal condition are satisfied. The main property of a counter-tag is that any initial state in which a plan is invalid features a counter-tag to the plan. For tag $ct = \langle c, t \rangle$, we write $[\![ ct ]\!]$ the set of initial states $i$ that satisfy this tag: $i \cap c = t$; furthermore, given a set $C$ of tags, we write $[\![ C ]\!]$ the set of initial states that satisfy at least one of these tags: $[\![ C ]\!] = \bigcup_{ct \in C} [\![ ct ]\!]$.

**Theorem 1** *The set of states in which candidate plan $\pi$ is invalid is $[\![ CTags(\pi) ]\!]$.*

Proof sketch of Theorem 1 is in the Appendix. Theorem 1 characterises the states in which a candidate plan is invalid. Of particular interest is the fact that one does not need to enumerate all initial states.

### 4.1 Deterministic DNNF

The probability of a set of counter-tags selected in each iteration is hard to compute by simply summing up the probability of each initial state. To enhance efficiency, we use Deterministic Decomposable Negation Normal Form, d-DNNF (Darwiche and Marquis 2002), since it permits polynomial time implementations of computing probability (Darwiche 2001; Huang 2006).

d-DNNF is a language for propositional formulas that satisfies *decomposability* and *determinism* conditions. Decomposability guarantees that the variables within each conjunction are distinct, and do not overlap. Determinism guarantees that the disjuncts within any disjunction are logically contradictory – E.g., if $\phi_1 \vee \phi_2$ appears as a subformula, then we also know $\neg \phi_1 \vee \neg \phi_2$. d-DNNF enables computation of certain logically complex queries in polynomial time in the size of the formula.

Let $\varphi$ be a formula in d-DNNF over a set of Boolean ('chance') variables $X = vars(\varphi)$. Let $P : X \rightarrow [0, 1]$ be the *a-priori* probability of each variable to be *true* ($P$ is independent for each variable). Then the probability that a random assignment of the variables of $X$ satisfies $\varphi$, written $count(\varphi)$, can be computed in polynomial time (Darwiche 2001; Darwiche and Marquis 2002; Huang 2006).

### 4.2 Computing the Probability of a CTS

In this section, we show how to compute the probability of the set of states $[\![ C ]\!]$ using d-DNNFs.

Importantly, in order to be able to use symbolic tools to reason about the probability of a set of states, we need the distribution function $I$ to be represented symbolically.

We assume that the set $F$ of facts is partitioned into $m$ subsets $F_1, \ldots, F_m$ (i.e., $F = F_1 \cup \cdots \cup F_m$ and for all $j, k$, $F_j \cap F_k = \emptyset$). We further assume that each subset $F_j$ is associated with a *oneof probability distribution* $M_j$ of $F_j$, i.e., a probability distribution over some subsets of $F_j$. The probability of an initial state $i$ is then the product of the probabilities of the intersections of this state with each $F_j$: $I(s) = M_1(i \cap F_1) \times \cdots \times M_m(i \cap F_m)$.

In our running example, the set of facts is partitioned into $F_x = \{x_1, x_2, x_3\}$ and $F_y = \{y_1, y_2, y_3\}$ and the oneof probability distributions are:

$$M_x = \begin{cases} \{x_1\} & \mapsto & 0.2 \\ \{x_2\} & \mapsto & 0.7 \\ \{x_3\} & \mapsto & 0.1 \end{cases} \quad M_y = \begin{cases} \{y_1\} & \mapsto & 0.2 \\ \{y_2\} & \mapsto & 0.7 \\ \{y_3\} & \mapsto & 0.1 \end{cases}$$

In this example, the subsets $F_x$ and $F_y$ both match a context, but this is not true in general.

Consider a oneof probability distribution $M_j$ over $F_j$, and let us write $\{i_1, \ldots, i_k\}$ the domain of $M_j$ (i.e., $M_j(i_p)$ is defined for all $p \in \{1, \ldots, k\}$). We create a set $X_j = \{\chi_{j,1}, \ldots, \chi_{j,k}\}$ of $k$ *chance variables* and, for each index $p$, a formula that associates $\chi_{j,p}$ with $i_p$ as follows:

$$\varphi_{j,p} = \left( \chi_{j,p} \wedge \bigwedge_{q < p} \neg \chi_{j,q} \right) \to \left( \bigwedge_{f \in i_p} f \wedge \bigwedge_{f \in F_j \setminus i_p} \neg f \right).$$

Finally, we write $\varphi_j = \bigwedge_{p \in \{1, \ldots, k\}} \varphi_{j,p}$.

In other words, for any Boolean assignment $\alpha$ of $F \cup X_j$ that satisfies $\varphi_j$, if $\chi_{j,\ell}$ is the chance variable with the lowest $\ell$-index such that $\alpha[\chi_{j,\ell}]$ is *true*, then the state $i$ represented by $\alpha$ satisfies $i \cap F_j = i_j$. Finally, we need to set the appropriate probabilities of the chance variables; these probabilities are calculated as follows:

$$\chi_p \mapsto M_j(i_p) / \left( 1 - \sum_{q < p} M_j(i_q) \right).$$

Given a formula $\varphi$ that represents a set of states (such as the set $[\![C]\!]$), the probability of the initial state satisfying $\varphi$ can be computed via:

$$count(\exists F. \varphi \wedge \bigwedge_{j \in \{1, \ldots, m\}} \varphi_j). \tag{1}$$

In our running example, the value of chance variables under $M_x$ and $M_y$ are $\{\chi_{x,1} = 0.2, \chi_{x,2} = 0.7/(1 - 0.2) = 0.875, \chi_{x,3} = 1\}$, $\{\chi_{y,1} = 0.2, \chi_{y,2} = 0.7/(1 - 0.2) = 0.875, \chi_{y,3} = 1\}$, respectively. The formulas associated with these chance variables are:

$$\begin{cases} x_1 \to \chi_{x1} \\ x_2 \to \chi_{x2} \wedge \neg \chi_{x1} \\ x_3 \to \chi_{x3} \wedge \neg \chi_{x1} \wedge \neg \chi_{x1} \end{cases} \quad \begin{cases} y_1 \to \chi_{y1} \\ y_2 \to \chi_{y2} \wedge \neg \chi_{y1} \\ y_3 \to \chi_{y3} \wedge \neg \chi_{y1} \wedge \neg \chi_{y1} \end{cases}$$

This six-clause formula $\varphi$ has the property that each assignment of chance variables satisfying the formula will result in a unique initial state. In general, the probability of this

state is determined by the production of values of chance variables assigned as *true*.

Given a set of counter-tags $C$, the initial states with chance variables represented by $C$ can be expressed as:

$$\varphi \wedge \bigvee_{c \in C} c$$

By compiling the above formula into d-DNNF, the probability of initial states represented by a set of counter-tags can be computed via the function *count*.

## 5 Computing a Candidate Plan

In this section, we show how to generate a classical planning whose solution set is exactly all the plans that are valid for a given set of CTS. For this purpose, we first show two operators that take as input two classical planning problems and compute a problem whose solution set is the union or the conjunctions of that of the input problems. Then we show how to use these operators.

### 5.1 Operations on Planning Problems

The following operations assume that the planning problems have disjoint sets of facts. If this is not the case, we assume that the facts are renamed to satisfy this condition.

**Definition 3** *Let $\mathcal{P}_1$ and $\mathcal{P}_2$ be two classical planning problems with $F_1 \cap F_2 = \emptyset$ and $\{name(a) \mid a \in A_1\} = \{name(a) \mid a \in A_2\}$. The* conjunction *of $\mathcal{P}_1$ and $\mathcal{P}_2$ is the classical planning problem $\mathcal{P} = \mathcal{P}_1 \otimes \mathcal{P}_2$ where*

- $F = F_1 \cup F_2$, $i = i_1 \cup i_2$, $G = G_1 \wedge G_2$, and
- $A = \{a_1 \otimes a_2 \mid a_1 \in A_1 \wedge a_2 \in A_2 \wedge name(a_1) = name(a_2)\}$ *where $a = a_1 \otimes a_2$ is defined by*
  - $name(a) = name(a_1)$,
  - $pre(a) = pre(a_1) \wedge pre(a_2)$, *and*
  - $eff(a) = eff(a_1) \cup eff(a_2)$.

$\otimes$ is called 'conjunction' because of the following result:

**Lemma 1** *Let $\mathcal{P}_1$ and $\mathcal{P}_2$ be two classical planning problems. The following holds: $\Pi(\mathcal{P}_1 \otimes \mathcal{P}_2) = \Pi(\mathcal{P}_1) \cap \Pi(\mathcal{P}_2)$*

Proof sketch of Lemma 1 is in the Appendix. It is easy to extend Lemma 1 to multiple problems: $\Pi(\mathcal{P}_1 \otimes \cdots \otimes \mathcal{P}_n) = \Pi(\mathcal{P}_1) \cap \cdots \cap \Pi(\mathcal{P}_n)$.

**Definition 4** *Let $\mathcal{P}_1$ and $\mathcal{P}_2$ be two classical planning problems with $F_1 \cap F_2 = \emptyset$ and $\{name(a) \mid a \in A_1\} = \{name(a) \mid a \in A_2\}$. The* disjunction *of $\mathcal{P}_1$ and $\mathcal{P}_2$ is the classical planning problem $\mathcal{P} = \mathcal{P}_1 \oplus \mathcal{P}_2 = \langle F, A, I, G \rangle$ where*

- $F = F_1 \cup F_2$, $i = i_1 \cup i_2$, $G = G_1 \vee G_2$, and
- $A = \{a_1 \oplus a_2 \mid a_1 \in A_1 \wedge a_2 \in A_2 \wedge name(a_1) = name(a_2)\}$ *where $a = a_1 \oplus a_2$ is defined by*
  - $name(a) = name(a_1)$,
  - $pre(a) = pre(a_1) \vee pre(a_2)$, *and*
  - $eff(a) = eff(a_1) \cup eff(a_2)$.

Unlike the conjunction, the disjunction requires an addition condition in order to get the expected behaviour.

**Lemma 2** *Let $\mathcal{P}_1$ and $\mathcal{P}_2$ be two classical planning problems. If $\mathcal{P}_1$ and $\mathcal{P}_2$ are such that their action preconditions are trivial ($\forall j \in \{1,2\}. \forall a \in A_j. \ pre(a) = true$), then the following holds: $\Pi(\mathcal{P}_1 \oplus \mathcal{P}_2) = \Pi(\mathcal{P}_1) \cup \Pi(\mathcal{P}_2)$*

Proof sketch is in the Appendix. It is easy to extend Lemma 2 to multiple projected problems: $\Pi(\mathcal{P}'_1 \oplus \cdots \oplus \mathcal{P}'_n) = \Pi(\mathcal{P}'_1) \cup \cdots \cup \Pi(\mathcal{P}'_n)$.

For example, consider the candidate plan $\pi = \text{D}$, and $\{x_3\}$ and $\{y_3\}$ are counter-tags to $\pi$. Since $\{x_3\}$ belongs to $c_1$ and $\{y_3\}$ belongs to $c_2$, we can have $\mathcal{P}'_x \oplus \mathcal{P}'_y$ in which the goal is $x_2 \vee y_2$.

To use the disjunction operator, we need planning problems with trivial preconditions. This is achieved with the trivialisation operation. Essentially, fact $\zeta$ is added to the problem. This fact is initially true and must remain true. When an action is taken whose original precondition is not satisfied, then $\zeta$ is made false through a conditional effect.

**Definition 5** *The* trivialisation *of planning problem $\mathcal{P}$ is the planning problem $Triv(\mathcal{P}) = \mathcal{P}'$ defined by*

- $F' = F \cup \{\zeta\}$ *where $\zeta \notin F$ is a new fact;*
- $A = \{Triv(a) \mid a \in A\}$ *where $Triv(a) = \langle name(a), true, eff(a) \cup \langle \neg pre(a), \emptyset, \{\zeta\}\rangle\rangle$;*
- $I' = \{(i \cup \{\zeta\} \mapsto v) \mid (i \mapsto v) \in I\} \cup \{(i \mapsto 0) \mid i \subseteq F\}$;
- $G' = G \wedge \zeta$;
- $\tau' = \tau$.

**Lemma 3** *Let $\mathcal{P}$ be a planning problem. The following holds: $\Pi(\mathcal{P}) = \Pi(Triv(\mathcal{P}))$.*

Prove sketch of Lemma 3 is in the Appendix.

### 5.2 Putting it All Together

Given a set $T$ of CTS, we show how to generate a new planning problem (*composite problem*) such that all the valid plans are exactly those that the counter-tag sets are not counter-examples to:

$$P_B = P_{T_1} \otimes \cdots \otimes P_{T_n}$$

where $B = \{T_1, \ldots, T_n\}$ and

$$P_{T_i} = P_{c_1} \oplus \cdots \oplus P_{c_m}$$

where $T_k = \{c_1, \ldots, c_m\}$ and $P_{c_k}$ is the trivialisation of the projection of $P$ over $c_k$. The composite problem is a classical planning problem.

At each iteration, OURALGO must generate a new counter-tag set (or terminate). The number of counter-tag sets is finite: the algorithm must terminate.

## 6 Conjunctive Goals

Our general approach requires to compute candidate plans from the set $\Pi(P_1) \cap \cdots \cap \Pi(P_n)$ where $\Pi(P_j) = \Pi(P_{ij}) \cup \cdots \cup \Pi(P_{jk_j})$. This is difficult because it leads to planning problems with disjunctive goals (as hinted at by the union operator $\cup$).

Notice that any candidate plan then belongs to the set $\Pi(P_H) = \Pi(P_{1h_1}) \cap \cdots \cap \Pi(P_{nh_n})$ where for all $j$, $h_j \in \{1, \ldots, k_j\}$, i.e., $H = \{h_1, \ldots, h_n\}$ forms a *hitting*

---

Algorithm 2: OURALGOHIT

1: **Input:** PCP problem $\mathcal{P}$
2: **Output:** a plan $\pi$ for $\mathcal{P}$, or UNSAT
3: $B := \emptyset$ ▷ a set of counter-tags
4: $\pi := \varepsilon$ ▷ candidate plan
5: **loop**
6:     **while** $\pi = \bot$ **do**
7:         $H :=$ unique_min_hit($B$)
8:         **if** $H = \bot$ **then return** no plan
9:         $\pi :=$ compute_candidate_plan($\mathcal{P}, H$)
10:    $CTS :=$ compute_CTS($\mathcal{P}, \pi$)
11:    **if** $CTS = \bot$ **then return** $\pi$
12:    $B := B \cup \{CTS\}$

---

*set* (Slaney 2014) of the set of counter-tag sets $B$. If $H$ were known in advance, finding this plan would be much faster because it only involves conjunctive goals.

To leverage this property, we propose to first search over the set of minimal hitting sets of $B$. This is summarised on Alg. 2. OURALGOHIT chooses a hitting set $H$, searches for a plan in $P_H$ and, if it fails, moves to the next hitting set. Searching for a candidate plan involves an extra loop to iterate over the hitting sets. Method unique_min_hit returns a different hitting set at each iteration. If all hitting sets fail to lead to a plan (Line 7 where the set of hitting sets has been exhausted), then there is no plan.

## 7 Experiments

We compared four different probability thresholds: $\tau \in \{0.99, 0.90, 0.75, 0.5\}$. The classical planners used in our implementations are the Fast Forward (FF) (Hoffmann 2001) and Madagascar [3] (Rintanen 2014).

Both OURALGO and OURALGOHIT are tested in the experiments. In Table 2 and Table 3, column FF-OUR corresponds to FF used in OURALGO; column FF-HIT corresponds to FF used in OURALGOHIT; column MAD-OUR corresponds to Madagascar used in OURALGO; column MAD-HIT corresponds to Madagascar used in OURALGO-HIT. As FF times out for nearly all problems at a probability threshold of 0.5, we only executed Madagascar under the 0.5 probability threshold. We did also experiment using classical planners based on the Fast Downward System (Helmert 2006) but do not include those results due to poor runtime performance in the preprocessing component of that tool on composite problems we encounter in our study. Due to space limitations, not all experimental results are presented in the tables. Full results are in the supplementary material.

We ran experiments over the set of benchmarks from Zhang, Grastien, and Scala (2020). We modified these CP problem benchmarks by incorporating a uniform distribution into the initial states. The benchmark set contains 6 domains: BOMB, COINS, DISPOSE, ONEDISPOSE, LOOK-GRAB, and UTS. Because OURALGO and OURALGOHIT randomly select counter-tags in each iteration, to mitigate

---

[3]We ran algorithm C with R=1.2, checking horizons from 0 to 50, using $\exists$-step -semantics.

the impact of randomness on the experiment, each instance was solved nine times, and we report the median result. Timeout was set to 1800 secs. The experiment was conducted on an Intel(R) Core(TM) i7-7700 CPU with 8 cores and 16 GB memory. The experiment did not include a comparison with other PCP planners, such as probabilistic-FF and POND, as these planners were developed decades ago and would not compile and/or run correctly on systems available to us.

The probability threshold 0.99 problems perform best over all the other probability threshold. This is because in each iteration the probability of counter-tags over 0.01 is enough to be used to search next candidate plan. In some problems, such as DISPOSE p-8-1 and UTS p7, there is only one context and the probability of any single initial state is more than 0.01, so only one counter-tag is needed in each iteration. This makes the composite problem become much easier to be solved as there is no disjunctive goal. Also, because of this reason, the benefit of hitting set strategy is not exhibited. For example, solving DISPOSE p-8-1 problem takes almost the same time by FF-OUR and FF-HIT.

As the probability threshold $\tau$ decreases, the performance of both OURALGO and OURALGOHIT gradually deteriorates. This phenomenon is due to two reasons. First, as the number of contexts (intuitively, dimensionality) of a problem becomes large, so too does the typical length of clauses in the goal of composite problem. Thus, for low values of $\tau$, we will oftentimes have many and long clauses in the goal of composite problem. The fact is that there are many correlates with the number of iterations required to solve the problem, and therefore the runtime. Moreover, for lower values of $\tau$ the number of clauses in the goal of the composite problem being posed is typically high. In practice, for low values of $\tau$ in problems with many initial states, we require many iterations of our algorithm to find a valid plan of PCP. The fact is that there are many correlates with the number of iterations required to solve the problem, and therefore the runtime. Both the length and number of clauses impact the performance of the base classical procedure, thus also negatively impacts overall performance.

The algorithms we have described have a performance profile that is quite different to that of incumbent algorithms. Specifically, POND and probabilistic-FF exhibit their fastest performance with lower values of $\tau$, with published performance degrading significantly with values of $\tau$ close to 1. (Domshlak and Hoffmann 2007). This is an interesting phenomenon, as our algorithm is good at handling high probability thresholds because it uses counter-tags for plan searches, and the higher the probability threshold, the fewer counter-tags are required, i.e., intuitively, as $\tau$ approaches 1 the problem becomes more-and-more like a typical fully conformant problem. In doing so, our algorithm effectively mitigates the inherent weaknesses found in other algorithms.

OURALGOHIT outperforms OURALGO significantly using both FF and Madagascar as the subplanner, especially when $\tau < .99$. This is what we expected, as there are multiple counter-tags in each iteration for all the problems with $\tau < .99$, and hitting set helps classical planners in selecting which counter-tag needs to be satisfied in the subse-

quent candidate plan. This can be evidenced by the ratio ($r$) of classical planner runtime to the total runtime (shown on the supplementary material). When $\tau < .99$, the value of $r$ in OURALGOHIT is significantly lower than in OURALGO. For example, when solving BOMB p20-10 with $\tau < .99$ using Madagascar, the $r$ value in OURALGO is over 50%, while in OURALGOHIT, it is less than 20%. In principle, there are various strategies for choosing a hitting set, instead of simply choosing smallest one. For example, our algorithm could heuristically take the initial state distribution into account when selecting hitting sets.

We observe that Madagascar performs better in solving problems with a small $\tau$ compared to FF. For example, when using OURALGO with a probability threshold of 0.75, MAD-OUR takes 14.8s to solve COINS p10, whereas FF-OUR times out. Another example is using OURALGOHIT with a probability threshold of 0.5, where MAD-HIT successfully solves UTS p9 and p20, whereas FF-HIT times out. This outcome implies that, in our algorithm, various classical planners can be chosen for different types of problems to enhance search efficiency. This underscores the flexibility of our algorithm.

## 8   Conclusion

In this article we present a counter-example based approach to solving PCP problems with uncertain initial states. In OURALGO, a PCP problem is projected onto contexts, and a set of counter-tags is then selected whose probability is computed according to a d-DNNF representation. OURALGO iteratively defines a composite problem (classical planning problem) that, at the limit, faithfully represents some underlying PCP of interest. At each iteration a classical plan is computed for the current composite problem at hand, or otherwise it is shown to be unsolvable. If the latter is the case, the underlying PCP has no solution. Otherwise, either a plan for the PCP is found corresponding directly to the classical plan, or that plan is used as a basis for refining the classical problem for a subsequent iteration. Composite problems with disjunctive goals are hard to solve using classical planning systems, such as Mad and FF. Hence, we developed OURALGOHIT, introducing a hitting set to assist the classical planner in selecting subgoals that can be satisfied by candidate plans.

From experimental results, we confirm the effective performance of both OURALGO and OURALGOHIT in solving PCP problems, and OURALGOHIT outperforms OURALGO. Our approach excels in addressing PCP problems with a high probability threshold, while other well-known PCP planners like probabilistic-FF and POND are good at solving PCP problems with a low probability threshold. This feature enables our approach to compensate for the weaknesses of current PCP planners, providing the planning community with enhanced options for solving PCP problems with high probabilities.

Another advantage of our approach is flexibility, allowing us to choose the classical planner based on the type of PCP problems and the features of different classical planners.

In the future, we can further improve OURALGOHIT by using better strategies in selecting hitting sets.

| Domain | Instance | $\tau = .99$ Runtime(s) / Iterations | | | | $\tau = .90$ Runtime(s) / Iterations | | | |
|---|---|---|---|---|---|---|---|---|---|
| | | FF-Our | FF-Hit | Mad-Our | Mad-Hit | FF-Our | FF-Hit | Mad-Our | Mad-Hit |
| Bomb | p20-5 | 4.19/21 | 4.32/12 | 4.85/21 | 5.11/13 | - | 3.49/5 | 63.1/50 | 3.61/6 |
| Bomb | p20-10 | 4.18/21 | 3.53/13 | 5/21 | 4.82/12 | - | 2.99/5 | 79.6/109 | 3.26/6 |
| Bomb | p20-20 | 3.23/20 | 2.88/9 | 5.08/18 | 3.16/6 | - | 2.24/5 | - | 2.49/4 |
| Bomb | p100-10 | - | - | - | 1093/30 | - | - | - | - |
| Bomb | p100-100 | - | 394/30 | - | - | - | - | - | - |
| Coins | p10 | 1.34/17 | 1.48/17 | 2.43/17 | 2.33/17 | 1.25/17 | 1.31/17 | 2.23/17 | 2.4/16 |
| Coins | p12 | 15.1/49 | 15/49 | 46.9/49 | 47.5/49 | 14.7/49 | 14.9/49 | 54.1/49 | 50.9/49 |
| Coins | p16 | 33.1/49 | 33.2/49 | - | - | 32.3/49 | 35.4/49 | - | - |
| Coins | p20 | 32.4/49 | 35/49 | - | - | 32.7/49 | 34.2/49 | - | - |
| Dispose | p-4-1 | 1.92/17 | 1.95/17 | 31.4/17 | 35.7/17 | - | 26.3/121 | - | 236/83 |
| Dispose | p-4-2 | 6.9/33 | 7.74/33 | 125/33 | 127/33 | - | 357/497 | - | - |
| Dispose | p-4-3 | 21.4/49 | 21.8/49 | 280/49 | 330/48 | - | - | - | - |
| Dispose | p-8-1 | 691/65 | 705/65 | - | - | - | - | - | - |
| LookGrab | p-4-1-1 | 0.87/5 | 1.04/4 | 1.27/4 | 1.48/6 | 1.15/5 | 1.19/5 | 13.5/4 | 0.86/3 |
| LookGrab | p-4-1-2 | 0.48/2 | 0.52/2 | 0.57/2 | 0.55/2 | 0.51/2 | 0.52/2 | 0.63/2 | 0.6/2 |
| LookGrab | p-4-1-3 | 0.51/2 | 0.52/2 | 0.59/2 | 0.56/2 | 0.53/2 | 0.52/2 | 0.63/2 | 0.61/2 |
| LookGrab | p-8-1-2 | - | - | - | - | - | - | - | - |
| OneDispose | p-2-2 | 0.68/7 | 0.67/7 | 1.1/9 | 1.14/9 | - | 1/8 | 274/23 | 1.5/9 |
| OneDispose | p-2-3 | 2.84/8 | 3/8 | 6.2/16 | 6.01/18 | - | 165/16 | - | 330/31 |
| OneDispose | p-4-2 | - | 153/40 | - | - | - | - | - | - |
| Uts | p7 | 2.03/15 | 1.81/15 | 90.1/13 | 67.1/13 | - | 22.5/92 | - | 89.8/20 |
| Uts | p8 | 2.71/17 | 2.79/17 | 269/15 | 227/14 | - | 41.3/121 | - | 199/21 |
| Uts | p9 | 3.3/19 | 3.74/19 | - | - | - | 68/154 | - | - |
| Uts | p20 | 4.89/21 | 5.25/21 | - | - | - | 104/191 | - | - |
| Uts | p40 | 48.4/41 | 50/41 | - | - | - | - | - | - |

Table 2: Performance of OurAlgo and OurAlgoHit when probability threshold $\tau$ is .99 and .9. This table presents the runtime and the number of iterations. FF-Our is FF used in OurAlgo; FF-Hit is FF used in OurAlgoHit; Mad-Our is Madagascar used in OurAlgo; Mad-Hit is Madagascar used in OurAlgoHit. "-" means TIMEOUT.

| Domain | Instance | $\tau = .75$ Runtime(s) / Iterations | | | | $\tau = .50$ Runtime(s) / Iterations | |
|---|---|---|---|---|---|---|---|
| | | FF-Our | FF-Hit | Mad-Our | Mad-Hit | Mad-Our | Mad-Hit |
| Bomb | p20-5 | - | 7.48/9 | - | 6.7/9 | - | 1.91/2 |
| Bomb | p20-10 | - | 3.38/5 | 82.5/57 | 4.86/7 | 5.31/3 | 1.73/2 |
| Bomb | p20-20 | - | 4.14/7 | 2.38/3 | 3.62/4 | 3.22/2 | 1.85/2 |
| Bomb | p100-10 | - | - | - | - | - | - |
| Bomb | p100-100 | - | - | - | - | - | - |
| Coins | p10 | - | 17.9/121 | 14.8/27 | 11/60 | 16.3/32 | 18.8/95 |
| Coins | p12 | - | - | - | - | - | - |
| Coins | p16 | - | - | - | - | - | - |
| Coins | p20 | - | - | - | - | - | - |
| Dispose | p-4-1 | - | - | - | 214/186 | - | 42.2/71 |
| Dispose | p-4-2 | - | - | - | - | - | - |
| Dispose | p-4-3 | - | - | - | - | - | - |
| Dispose | p-8-1 | - | - | - | - | - | - |
| LookGrab | p-4-1-1 | 1.75/5 | 1.17/4 | 35.4/5 | 1.31/4 | 2.35/3 | 0.82/2 |
| LookGrab | p-4-1-2 | 0.6/2 | 0.59/2 | 0.92/2 | 0.63/2 | 1.12/2 | 0.75/2 |
| LookGrab | p-4-1-3 | 0.64/2 | 0.59/2 | 1.09/2 | 0.64/2 | 1.28/2 | 0.74/2 |
| LookGrab | p-8-1-2 | - | - | - | 190/15 | - | 101/9 |
| OneDispose | p-2-2 | - | 2.48/14 | - | 4.01/20 | 84.2/21 | 5.32/6 |
| OneDispose | p-2-3 | - | - | - | - | - | - |
| OneDispose | p-4-2 | - | - | - | - | - | - |
| Uts | p7 | - | 435/1002 | 827/4 | 65.5/22 | 234/2 | 31.3/12 |
| Uts | p8 | - | - | - | 97.1/17 | - | 115/23 |
| Uts | p9 | - | - | - | 162/24 | - | 107/13 |
| Uts | p20 | - | - | - | - | - | 108/11 |
| Uts | p40 | - | - | - | - | - | - |

Table 3: Performance of OurAlgo and OurAlgoHit when probability threshold $\tau$ is .75 and .5.

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
