# OpenReview forum: "A Counter-Example Based Approach to Probabilistic Conformant Planning"
_icaps-conference.org/ICAPS/2024/Conference — ICAPS 2024_

### Official Review · Reviewer_PopZ · 2024-01-09

**Significance And Importance:** 2
**Soundness:** 2
**Novelty:** 3
**Clarity:** 2
**Overall Evaluation:** 2
**Confidence:** 4

**Weaknesses:**

0: Minor weaknesses requiring some work to be addressed for the paper to be accepted.

**Contributions Of The Paper:**

This work builds on the work of Scala & Grastien by using their counter-example based planner CPCES to synthesise probabilistic conformant planning solutions.
The counter-example based approach is novel and interesting when applied to this type of problem. In this sense, the paper is very relevant to ICAPS and an interesting piece of work.

**Ethical Considerations:**

(1) Not Applicable: The paper does not have any ethical considerations to address

**Nomination For Best Paper:**

No

**Questions For Authors:**

1. The authors focus on PCP with unknown initial states. What is the cost of extending the approach to ND actions?
2. Do you have some complexity results to show for the two algorithms?

**Reproducibility:**

3: Authors describe the implementation and domains in sufficient detail.

**Strengths Of The Paper:**

- Introduces a counter-example based approach for solving probabilistic conformant planning problems
- Algorithm seems to be sound and complete
- Variation of the algorithm using hitting sets accelerates generation of candidate plans
- Particularly suited for problems with a high probability threshold

**Weaknesses Of The Paper:**

My primary concern lies with the technical intricacies of the work. The paper's style vacillates between formal definitions and theorems, and a high-level, abstract exposition of the approach. I find this to be exceedingly perplexing, as the impreciseness of the definitions renders it challenging to assess the soundness of the methodology. On the other hand, the abundance of details obfuscates the "fil rouge" of the approach, which is a closed-loop algorithm, iteratively seeking a valid plan by computing a series of counterexamples. I propose adhering to a consistent style (such as a "formal" one) and maintaining it within the main body of the paper.

There are few flaws in the definitions and in the description of the approach. The clarity depends also in the order of the presentation. I suggest, for instance , to move sec. 6 before 5.2 (p. 6), or describe OURALGO after introducing CTS (p. 3).

The approach looks expensive as the set of candidate plans is generated incrementally until a plan with the desired probability is found. In the worst case, the CTS is made of all initial states for which the plan is invalid, which looks like proportional to the superset of 2^F. Some complexity result would be welcome to justify this particular approach.

Said that, I insist saying that this paper is interesting and relevant for the conference, even if, as it is now, it deserves further polishing.

# More important technical flaws

- Definitions must be well stated. For instance, the definition of subgoal, which is important,is vague but we encounter it all along the paper. I understood that a subgoal can be any goal or any precondition conjunct. More in general to the section 2: if you're using Palacions&Geffner's definition, please use the same terms as well. Here 'depends' means 'relevant'; 'contect' is 'relevant clause'; 'tag' is 'relevant initial clauses' (maybe?). In the present work, the author miss the transitivity of 'relevant' definition.
- p.3. "y_2 depends on y_3" is incorrect: y_2 depends on y_1 as well per D.
- p.3. "the context of a subgoal includes all the facts whose truth (value) in the initial state can influence the validity of the subgoal in any state" sounds a false statement. As defined, the context includes also facts that are preconditions of actions in the plan, not necessarily relevant for the initial state.
- p.4. another weakness in the formality of the writing is definition 1.
- - who are \phi_j?
- - if eff+ U eff- in c, then c \cap eff+ = eff+! I thus suggest to write directly the projection and get rid of the second part of the formula. Why the effects are in the context then?
- sec 5.2. Please define P, and c_k. I would rather write: "Let P be a planning problem and the set {P_{c_k}}_{c_k} the trivialisation of the projections of P over contexts c_k. It is clear that T_k is a collection of contexts s.t. B = {T_k}_0<k<n
- at the end of sec 5.2 you have a result relative to the termination of OURALGO placed here without any explaination. Better to say how the composite problem is made, and the counter-examples found, in order to solve the composition of the projections of the different problems over their contexts.
- Beginning of sec. 6 Not clear who are the different sets. you change the indexes from section to section, and it's really hard to follow. P_{jk_j} really puzzled me at this point. I really recommend to rewrite this section ,in order  to introduce you selection of counter-tags. It's here that the reader really misses a general descritpion of this version of OURALGO.

# Minor stuff

- p.1. A remark: in CP, 'weak' solutions are excluded
- p.1. to reach the destination G = (x_2, y_2)... and later specify for clarity that "its initial position, which can be anywhere in the grid".
- p.2. typo : "Second, it translates"
- p.3. "precondition y_2 - I.e.," is better rewritten "precondition y_2, i.e."
- p.3. typo: "Consider subgoal"
- p.3. I find the last sentence of sec. 2 really unclear: to apply L it is obvious that you must know its precondition, which is y_1, the vertical coordinate. Note that reasoning with 'merges' à la P&G, all but one vertical position would be needed, not all of them.
- p.3. ref to algo 1 when encountering OURALGO for the first time.
- It would be clearer if lines in algo 1 are linked during the example p. 3. E.g. the first candidate plan (line 4). Possibly move end of sec. 3 before the example.
- Algo 1. replace "explained" by "see" to save 2 lines.
- p.5. typo: to generate a classical planning problem whose solutions...
- In the results p.8, please empathise 'best' results using bold typeface.
- Still in the tables p.8, please give some indication of how difficult is to find the plan iteratively, e.g. indicating the size of the CTS.
- Def. 5, please define P' = {F', A', I', G'}
- Def. 5. typo: A' =
- Proof sketch of th.1.
- - "we can have" -> "we have" (it's a proof, we do not want hypotheses here)
- - definition of subgoal of \phi is missing, because we don't know what \phi is.
- - I would rather write simply: The plan starts in the initial state i. If the plan fails, by definition i is a counter-tag to the plan.

---

> ### Author Rebuttal · Authors · 2024-01-27
>
> Thanks to all the comments and interesting suggestions. Below we answer all questions in detail. Of course, all minor comments will be taken into account should the paper be accepted for presentation.
>
> Q1A.
> The authors of CPCES, Scala and Grastien, have provided some ideas in ICAPS-2021 (Scala and Grastien, 2021) about how to address problems with uncertain action effects through a counter-example based approach. Specifically, their algorithm translates a conformant planning problem to a non-deterministic finite automaton (NFA), and searches the counter-automaton in each iteration. We could use a similar approach; this would require us to introduce extra chance variables to model the uncertainty on the action effects, as proposed in J. Huang’s work (Huang, 2006). This is indeed one of our future research directions.
> Scala, E., & Grastien, A. (2021, May). Non-deterministic conformant
> planning using a counterexample-guided incremental compilation to
> classical planning. In Proceedings of the International Conference on
> Automated Planning and Scheduling (Vol. 31, pp. 299-307).
>
> Q2A.
> Remember that PCP is ExpSpace-hard.
>
> Both algorithms have the same number of iterations: $O(2^T)$ where $T$ is the number of tags.  Each iteration requires 1.
> to compute a candidate plan for an increasingly difficult classical
> planning problem and 2. to determine the counter-tags and the
> probability of success.  Part 2. is dominated by part 1 (polytime
> problems vs PSPACE-hard ones).  The size of the classical planning
> problem (number of facts) is $O(F \times T)$ where $T$ is the
> number of tags and $F$ the number of facts in the PCP.  For OurAlgo,
> the goal formula of the classical planning problem could be $O(i \times
> d \times c)$ where $i$ is the current iteration number (potentially
> $2^T$), $d$ is the number of disjunctions (i.e., "one of the tags in the
> CTS must be satisfied", potentially $O(2^T)$), and $c$ is the length of
> a tag ($O(T)$).  For OurAlgoHit, the size of the goal formula is $O(T * F)$.
> OurAlgoHit also requires, at each iteration, to solve a minimal-cardinality hitting set problem, which is NP-complete.

---

### Official Review · Reviewer_AjoV · 2024-01-22

**Significance And Importance:** 3
**Soundness:** 3
**Novelty:** 4
**Clarity:** 3
**Overall Evaluation:** 3
**Confidence:** 5

**Weaknesses:**

0: Minor weaknesses requiring some work to be addressed for the paper to be accepted.

**Contributions Of The Paper:**

This paper presents novel planning algorithms for Probabilistic Conformant problems that result from synthesizing existing work. The first of these is the Grastien & Scala CEGAR algorithm for conformant planning. The second is the seminal application of Knowledge Compilation techniques by J. Huang to conformant planning with probabilistic information on possible initial states (aka Probabilistic Conformat planning).

The main facts of the contributions of this paper (as I understand them) are:

1. The planning algorithm is iterative, generating a sequence of candidate plans $\pi_1$, $\pi_2$, $\pi_3$, … which are feasible (e.g. they do not violate any causal constraints) but may not guarantee goal reachability above a given threshold.
2. The algorithm terminates when  $\mathrm{P}_{success}(\pi_k) > \sigma$, where $\sigma \in (0, 1]$, for some $k > 1$.
3. The algorithm is claimed to be sound and complete:
    - The algorithm terminates within finite time when no causally valid plans exist that satisfy probability bounds.
4. A second algorithm, a refinement of the former, uses results from the theory on Hitting Set Problems to speed up the generation of plans $\pi_k$

**Ethical Considerations:**

(1) Not Applicable: The paper does not have any ethical considerations to address

**Nomination For Best Paper:**

No

**Questions For Authors:**

1/ In the section "A Counter-Example based Approach" it is noted that unsolvable states $s$ are not to be added to the sample. This is a reasonable proposition, but the development of scalable algorithms depends on the assumption that proving the unsolvability of a state can be done quickly and feasibly. This seems to rule out the possibility of using classic SAT algorithms for classical planning unless one further assumes that all possible states in an initial conformant state are solvable. Could the authors please clarify what is meant in this passage? Is the sampling algorithm supposed to access an oracle for "weeding out" unsolvable states?

2/ Lemma 1 needs to consider relevant assumptions for its proof. Namely, that is the case that every valid plan, for instance ${\cal P}_1$, is also a valid plan for ${\cal P}_2$. At the moment, this assumption is "hidden" by having them very much in the first paragraph of the Proof of Lemma 1. I also think this assumption is of great consequence. What compelling reason can the authors give to accept this assumption in the first place? From what I have understood, it could well be the case that instances have plans that use entirely different sets of actions, and more frequently, I suspect, plans where valid plans for either instance use the same actions, but the plan assigns them different indices, or they have different multiplicity (e. g. here I am thinking of plans as functions mapping actions to *sets* of natural numbers representing the plan steps).

3/ Accepting the assumption in question 2 (which is also required by the proof of Lemma 1) and having checked the proof for Lemma 2 in the appendix, I am unsure of why the notion of "precondition triviality" is necessary. In the proof, I can read that "Condition 2 is trivial because preconditions are trivial", but there are no signposting numbering conditions. It is not clear to me what is exactly trivial. Could the authors please provide more clarity on this?

**Reproducibility:**

2: Some details are missing, but the paper still appears to be replicable with some effort.

**Strengths Of The Paper:**

I comment on the strengths of the paper separately for each section of the paper.

# Computation of Counter-Tags

This section presents an interesting synthesis of Huang's work on one side and Palacios and Geffner's work on the other (note that the latter also explored the applications of knowledge compilation technologies into conformant planning [1]). The notion of introducing auxiliary stochastic (chance) variables for choosing which one-ofs/state invariants are active to select initial states is inspired and points to a deep connection between stochastic SAT/CSP theory [2] and Probabilistic Conformant Planning. This is not entirely surprising: the work by Littman and Majercik [3] on Stochastic SAT and applications into Contingent Planning identified and exploited computationally this connection.

I have found several issues with the structure and clarity of the section. Suggestions on how to improve the paper follow:

- The discussion of $d$-DNNF and introduction of the two relevant operators (projection and counting) needs to be in the background section. Also, the remark about computing the probability of random assignments to variables of $d$-DNNF formulas needs to be more clearly attributed or presented (I think the remark follows from Algorithm 1 in Huang’s paper, right?). The (composite) operator $count(\exists X. \varphi)$ also needs careful explanation.

- There is an unfortunate typo in the definition of $\varphi_{j,p}$: the range of the conjunction in the antecedent requires that $q < p$, instead of $q \neq p$. The typo is then repeated below when giving the probability map.

- It would be useful to make an effort to hand-hold readers so nobody misses that there is a chance variable for every *univariate* tag. Tags can be composites (via conjunction), but $d$-DNNF operations aggregate the probabilities.

- The one-of structures or state invariants, as well as the definitions of the probability maps $M_j$, should be part of the definition given for PCP instances.

# Computing a Candidate Plan

I find this section to be brilliant and a great contribution with ramifications beyond PCP (can generalized pattern database heuristics to planning problems be expressed using this algebra or a suitable modification? how does it relate to or can be used in other applications of CEGAR to classical planning? how about provenance analysis for actions chosen by conformant or contingent plans?). The authors propose an algebra for factoring conformant planning problems according to sets of selected tags (properties that partition the set of states). The operators given have a strong similarity with the natural join and union operators in the positive Relational Algebra [4]. On the other hand, the authors have moved the proofs of the lemmas (which are critical) to the Appendix, without providing any insights into the structure of the arguments provided to support the Lemmas in this Section.

# Techniques for exploiting the structure of conjunctive goals

A very brief section that leverages important insights from a paper from Jon Slaney I wasn’t aware of but seems to have informed advances in QBF satisfiability. Again, this is an unsurprising but deep connection due to conformant planning being on the second rung of Papadimitriou’s characterization of the polynomial hierarchy as alternating layers of existential and universal quantification for FO formulas (see [5]). My single suggestion in this section (applicable to the rest of the Algorithm environments in the paper) is to consider using a different font for procedures so the authors can write $\mathrm{UniqueMinHit}$ or $\mathsf{UniqueMinHit}$ rather than unique_min_hit.

# References

[1] H. Palacios and H. Geffner, “Mapping Conformant Planning into SAT Through Compilation and Projection”, https://link.springer.com/chapter/10.1007/11881216_33, and also, H. Palacios, B. Bonet, A. Darwiche and H. Geffner “Pruning conformant plans by counting models on compiled d-DNNF representations”, https://dl.acm.org/doi/10.5555/3037062.3037081

[2] M. L. Littman, S. M. Majercik, T. Pitassi "Stochastic Boolean Satisfiability”, https://link.springer.com/article/10.1023/A:1017584715408

[3] S. M. Majercik, M. L. Littman, “Contingent planning under uncertainty via stochastic satisfiability”, https://www.sciencedirect.com/science/article/pii/S000437020200379X?via%3Dihub

[4] Abiteboul et al. “Foundations of Databases”

[5] H. Turner, "Polynomial-length planning spans the polynomial hierarchy", JELIA, 2002

**Weaknesses Of The Paper:**

The ICAPS PC team prepared a useful modified AAAI style that numbers lines. This is a useful feature I wish the authors had used. The authors are lucky that there is no automated system rejecting submissions that do not comply with the publishing guidelines.

# Occasional issues with clarity

The discussion in the related work section sometimes comes across as cryptic. For instance:

- The discussion of POND mentions that this algorithm is sensitive to the “number of states” but not the “problem complexity”. This is unclear to me because of two reasons. First, only states reachable with positive probability matter, regardless of what is syntactically possible. Secondly, existing complexity results that are specific for probabilistic conformant planning do not exist as far as I know. All we have is the existing theory for conformant planning. Its relation to problems with probabilistic lower bounds for success has not been studied deeply or systematically in the literature.

- When discussing Probabilistic-FF limitations, the authors make a distinction when I do not think there is any to make, these issues carry over from Conformant-FF, and it all links back to the properties of the Relaxed Planning Graph (RPG). If valid conformant plans require multiple instances of the same action, these will not appear on the RPG analysis, and the lookahead search will be less informed and more like that in POND. Also, when the RPG captures only a subset of the actions in valid conformant plans, this may result in a large search effort too.

- The clarity of Grastien & Scala's CPCES discussion could be improved. For instance, it is unclear how a plan $\pi$ can “agree” with all (initial) states in a set. Maybe it is too early to get into such detail.

- There is a typo in the text: “Second, it treanslates” for “Second, it translates”

# Minor issues in Problem and Background definitions

This section's quality is fair, but there are a number of issues with the definitions that warrant some further attention. Please find below some recommendations to improve quality and clarity:

- It would be possibly useful to preface the section by noting that the definitions and assumptions therein follow from those in STRIPS. Doing so allows us to locate this work precisely and quickly amongst formulations in the literature on AI planning.

- The wording of the definition of states is unfortunate. To my reading, it suggests that states are subsets of “true facts”. Therefore, states cannot reliably be used to interpret formulas $\varphi$, which is necessary to define the applicability relation between states and actions.

- A concise and precise way to define the propositional language for formulas $\varphi$ is that of the set of formulas over atoms $F$ without disjunction *and* negation.

- The notion of “name” is not defined. I would invite the authors to consider them to be function symbols (drawn from a suitably defined first-order theory), which plans to interpret into (possibly empty) sets of positive integer numbers.

- The characterization of $I$ (a probability distribution) via its graph (the set of pairs formed by elements of the domain and range of the function) is a hard sell. If $I$ is a probability distribution mappings states $S$ to $[0,1]$, then initial states for classical planning instances are probability distributions $I(s) = 1$ if $s = s_0$, and $I(s) = 0$ for every $s \neq s_0$. The way $I$ is used suggests that $I$ is taken to be the *support of* a probability distribution (the set of elements of the domain mapping to a positive real number) rather than the *actual* probability function.

- “a condition the algorithm aims to achieve”: what algorithm is this? Conformant plans can be used to compute automata satisfying a specification, and I guess one could reasonably imagine that probabilistic automata could also be captured by PCP plans. Please review this statement, as it seems to be a bit off. Interpreting plans as programs is not supported by your definitions.

- Regarding decorated “names,” perhaps a better word instead of "decoration" is "parametrization". That is, super and sub-indices are taken to be parameters used to define each of the elements in the definition of instances.

- I would suggest using captioning and italics for the paragraphs where the discussion of the motivating example is continued. This would be helpful to visually separate definitions from illustrations of the concepts being defined.

- I do not agree with the statement that “a PCP problem with only one initial state $i$ is equivalent to a classical planning problem”. This statement is only true when $\tau = 1$, as observed in footnote 2.

- There is a typo in the text (e.g. “ssubgoal”).

# Poor signposting and missing definitions in "A Counter-Example based Approach"

I found this section's quality to be mixed. There are some gaps in the explanation of Algorithm 1, and I found some of the points in the discussion to not follow clearly from the definitions provided in the previous section. I list below my suggestions to improve or bridge the issues I perceived:

- The section starts strong with an insightful discussion of why the strategy followed by the CPCES algorithm is not a good one for PCP problems because having unsolvable states in the sampled set of initial states would drive the probability of success to 0. In effect, CPCES strategy is too conservative for the PCP setting. I would suggest being more proactive in the text to signpost this insight to readers rather than leaving it as an exercise for them.

- A key concept is not defined explicitly: that of the probability of tags $t$ (e.g., sets of states satisfying the property represented by the tag). Since all states in the tag are mutually exclusive this is just the result of computing $P(t) = \sum_{s \models t} I(s)$. Less clear is the computation of the conjunction of two or more tags, since a state $s$ with $I(s) > 0$ can satisfy several tags. This should be addressed in the previous section.

- Algorithm 1 and the example computation seem to diverge in one critical aspect: the role of tags being accumulated through iterations. The discussion in step 2 does not provide any intuitions of why the tag $x_3$ selected in iteration $i=1$ is not considered by the plan generated in iteration $i=2$, but in contrast, $x_1$ is taken into account. The pseudo-code of Algorithm 1 does not suggest that tags not in $B \setminus CTS_i$ are “optional”.

# Some aspects in the Experimental Design and Analysis of Experimental Results

The experiments are designed sensibly, handling the random perturbations in a reasonable way. Benchmark selection seems to follow the inertia of previous publications rather than being informed by the specifics of the proposed approach. For instance, a key issue to my mind is the requirement to handle conditional effects where conditions are disjunctions. This only occurs when actions in domains (sets of instances sharing actions) have more than 1 precondition. As a matter of fact, FF does not support disjunction on conditional effects, unless they are compiled away (somehow). Such a compiler is not trivial and would warrant a detailed explanation.

There should be also some explanation of what algorithms are being used to solve incrementally the hitting set problems, which I understand eventually represent a larger fraction of runtimes than solving classical planning problems. Also, as suggested by the discussion on heuristics for selecting hitting sets, runtimes, and coverage may be sensitive in very substantial ways to the specific hitting sets being selected.

I have some actionable suggestions for the authors:

- Rather than taking the instances in benchmarks for granted, try to generate instances according to some suitably defined parametrization that generates diverse sets of initial state distributions. Also, by exploring a broader range of instances (especially “small” ones), we would get more information about how the planners compare. To be honest, there are so many time-outs for FF that I would suggest a healthy dose of prudence when interpreting the results.

- Classifying instances/domains according to their potential runtimes is important. There are several ways to do that: for instance, using the notion of conformant width and structural complexity of preconditions and goals (and therefore of tags). The work on structural complexity by Palacios and Geffner should be (along with that of structural diversity) a major factor in the experimental design.

---

> ### Author Rebuttal · Authors · 2024-01-27
>
> Thanks to all the comments and interesting suggestions. Below we answer all questions in detail. Of course, all minor comments will be taken into account should the paper be accepted for presentation.
>
> Q1A.
> Your intuition is not entirely correct. In Conformant Planning, the proof that a candidate plan is invalid is a single counter-example/state; CPCES adds this state to the sample.
> In PCP, a proof that the plan is invalid is a *set of counter-states* with probability above $\tau$. Therefore, we should not add a single state to the sample but a set of states. We do not want to enumerate these states so we use sets of counter-tags instead.
>
> Q2A.
> Note that $\pi$ is `valid` for $P$ means $\pi \in \Pi(P)$, not $\pi \in Actions(P)*$.
>
> Our only assumption is that $P_1$ and $P_2$ have the same set of actions: the assumption is guaranteed because it’s a precondition of operation $\otimes$. The definitions (precond., effects) of the same action in $P_1$ and $P_2$ are different.
>
> Let us clarify with our running example. Let $P_1$ and $P_2$ be the projections on tags $x_1$ and $y_1$ respectively. Actions $D$ and $L$ are defined in $P_1$ and in $P_2$. In $P_1$, $D$ has obvious effects (moves to the left); in $P_2$, $D$ has no effect because $P_2$’s facts only mention the vertical position.
> You can see $P_1\otimes P_2$ as two agents at different initial states doing the same actions, and finally both of them reaching their goal state. Plan $LD$ is a solution for $P_1$ and a solution for $P_2$; therefore, it is a solution for $P_1 \otimes P_2$.
>
> Q3A.
> Consider the projections $P_1$ and $P_2$ of the running examples on $y_1$ and $y_2$. Consider the plan $L$. This plan is invalid for $P_1$ (it does not reach the goal $y=2$) and for $P_2$ (action $L$ is not applicable from $y=2$). So $L$ should not be a valid plan for $P_1 \oplus P_2$.
> Using Def. 4, $L$ is applicable in the initial state (because it is applicable for the first interpretation) and the goal is reached (because it is reached in the second interpretation). The trivialisation operation makes sure the validity is only verified in the final state.

---

### Official Review · Reviewer_Bs74 · 2024-01-24

**Significance And Importance:** 2
**Soundness:** 4
**Novelty:** 3
**Clarity:** 4
**Overall Evaluation:** 1
**Confidence:** 3

**Weaknesses:**

2: No major or minor weaknesses.

**Contributions Of The Paper:**

- Introduction of a counter-example-based method for probabilistic conformant planning (PCP), which incrementally refines candidate plans using counter-examples.
- Development of an algorithm that is both sound and complete.
- Variant of the algorithm using hitting sets for increased efficiency.
- Experimental validation demonstrating the effectiveness of the proposed method, especially in high probability threshold scenarios.

**Ethical Considerations:**

(1) Not Applicable: The paper does not have any ethical considerations to address

**Nomination For Best Paper:**

No

**Questions For Authors:**

1) How does the hitting set strategy specifically contribute to the acceleration of candidate plan generation?

2) You mention that the proposed approach involves encoding the problem using a propositional formula and using a d-DNNF to assess the probability of a set of counter-tags to a candidate plan. Can you further explain how this encoding and assessment process is carried out, and how it contributes to the overall soundness and completeness of the algorithm?

3) You mention that the probability of a set of counter-tags selected in each iteration is hard to compute by simply summing up the probability of each initial state. Could you provide more insights into the specific challenges (or complexities) involved in computing the probability of these counter-tags, and how the use of d-DNNF effectively addresses these challenges?

**Reproducibility:**

2: Some details are missing, but the paper still appears to be replicable with some effort.

**Strengths Of The Paper:**

The paper presents a new counter-example based method for addressing PCP problems. This approach incrementally generates candidate plans and identifies counter-examples, offering a new perspective on solving PCP problems. A proof of the soundness and completeness of the proposed algorithm is also provided. Finally, the experimental results demonstrate that the proposed method excels in scenarios with high probability thresholds, which is often challenging for traditional planners. Moreover, the paper introduces a variation of the algorithm that incorporates a hitting set strategy to expedite the generation of candidate plans, showcasing the adaptability of the approach.

**Weaknesses Of The Paper:**

- The paper acknowledges that the approach shows a deterioration in performance as the probability threshold decreases. This is attributed to the increase in the number of contexts and the length of clauses in the goal of the composite problem. The impact of these factors on the performance of the algorithm should be further addressed. The algorithm faces challenges in solving problems with many initial states and low probability thresholds, but this is not thoroughly discussed.

- The benefit of the hitting set strategy is not exhibited in certain problems, and potential refinements or alternative approaches in such scenarios should be explored. Additionally, the complexity of computing minimal hitting sets should be in the second polynomial hierarchy, if I'm not mistaken. Thus, computing such sets is computationally hard and expensive. It is not clear in the paper how the hitting set approach actually speeds up the algorithm. This should be discussed.

- The paper does not provide publicly available code and domains, which is essential for reproducibility.

---

> ### Author Rebuttal · Authors · 2024-01-27
>
> Thanks to all the comments and interesting suggestions. Below we answer all questions in detail. Of course, all minor comments will be taken into account should the paper be accepted for presentation.
>
> Q1A
> In OurAlgo, the goal is a conjunction of disjunctions, for example `(A or B or C) and (A or C or F) and (B or G)`. These problems are hard to solve for heuristic-search planners (FF) and SAT-based planners (Madagascar) because the goal does not guide the search. We compute a minimal-cardinality hitting set, for instance `A and G`, and solve the problem with this conjunction as the goal. Planners are much faster with this type of goal.
>
> Q2A
> Our algorithm builds a d-DNNF that represents the probability of the initial state.  This probability is defined in the PDDL file through one-ofs. For instance, in BombInTheToilet with 3 bombs, if there is only one armed bomb, the oneof will specify {armed_b1: 1/3; armed_b2: 1/3; armed_b3: 1/3}. We create three chance variables A=1/3, B=1/2, C=1, and define the DNNF as: (A -> armed_b1) /\ (!A /\ B <-> armed_b2) /\ (!A /\ !B /\ C <-> armed_b3).
>
> Then, given a formula $\varphi$ that represents a set $S$ of states, $count(DNNF \land \varphi)$ is the probability of $S$.
>
> The probability of a d-DNNF is computed by replacing the leaves by the chance variable values, and `and` and `or` nodes with `*` and `+` (see Algorithm 1, Huang, 2006). The correctness of using d-DNNF to compute probability has been proved by (Darwiche 2001; Darwiche and Marquis 2002).
>
> Q3A
> The probability of a subset of tags cannot be computed solely from the probability of each individual tag because the tags are not pairwise independent. Thus, one generally needs to compute the sum of the individual probabilities (which is impractical). d-DNNFs (like BDDs) are compact representations of sets of states whose probability can be computed in linear time. We refer the reviewer to the literature on the use of BDDs for planning and model-checking for the benefits and pitfalls in these methods.
>
> In this work, we used the library DSHARP (Muise et, al. 2012) to compile the formula to a d-DNNF. DSHARP is able to compile a CNF formula to a d-DNNF based on sharpSAT (M Thurley, 2006).

---

### Meta-Review · Area_Chair_ucch · 2024-02-05

**Recommendation:** Accept (Oral)
**Confidence:** 4

**Metareview:**

The reviewers were satisfied with the rebuttal, and we all converged on an "accept" recommendation.

The discussion for the paper largely centered around reacting to the rebuttal. Two of the reviewers wrote answers to the rebuttal that were specifically intended as answers to the authors. Because the authors won't be able to see these comments, I am including them here.

*Answer by Reviewer Bs74 to the rebuttal:*

Thank you for the detailed response. I don't have any concerns about the paper now, but I do have the following suggestion that came to mind.

Explainability: A good application of this work that may be worth discussing and/or exploring, which I think will definitely increase its impact, is explainability. Specifically, the fact that you use logic to represent PCP problems makes it a perfect candidate for generating explanations about these problems. For example, given your set of formulas (the d-DNNF), you can find which formulas are responsible for which particular behavior and use them to explain it. Indeed, the method of computing minimal hitting sets (that you also utilize )has been explored in the context of explanation generation in explainable planning (particularly, model reconciliation), where the planning models are represented in logic [1]. You can easily see the parallels between these two works.

[1] S.L Vasileiou, A. Previti, W. Yeoh, "On Exploiting Hitting Sets for Model Reconciliation", AAAI 2021

*Answer by Reviewer AjoV to the rebuttal:*

> Q1A. Your intuition is not entirely correct. In Conformant Planning, the proof that a candidate plan is invalid is a single
> counter-example/state; CPCES adds this state to the sample. In PCP, a proof that the plan is invalid is a set of counter-states with
> probability above . Therefore, we should not add a single state to the sample but a set of states. We do not want to enumerate these states
> so we use sets of counter-tags instead.

Okay, this does clarify question. Please make sure that the above is conveyed in the paper as crisply and precisely as above.

Regarding the answer to my second question above

> Note that pi is valid for P means that pi in Pi(P) not pi in Actions(P)*

The difference is noted but my comment refers to the fact that Pi(P) \subseteq Actions(P)*. There is no assumption in the paper to restrict the structure of plans when it comes to the number of times actions appear in them.

> Our only assumption is that P_1 and P_2 have the same set of actions; the assumption is
> guaranteed because it is a precondition of operator \otimes. The definitions (precond. effects)
> of the same action in P_1 and P_2 are different.

Thank you for the patient comment. I have realised that I did not take into account the clause in Definition 3 that goes F_1 \cap F_2 = \emptyset. Indeed, you can have a valid plan for P_1 that has actions with trivial preconditions and effects, this plan can indeed be executable as trivial actions a with pre(a) = top and eff(a) = \emptyset are (1) applicable in any state, and (2) they give rise to transitions (s, a, s') for any pair of s, s' in S. Optimal plans for P_1 need not be optimal plans for P_2 but that is not an issue.

Regarding my third question

> Consider the projections P_1 and P_2 of running examples on y_1 and y_2. Consider the plan L. This plan
> is invalid for P_1 (it does not reach the goal y=2) and for P_2 (action L is not applicable from y=2). So L should
> not be a valid plan for P_1 \oplus P_2. Using Def. 4. L is applicable in the initial state (because it is applicable
> for the first interpretation) and the goal is reached (because its reached in the second interpretation). The
> trivialisation operation makes sure the validity is only verified in the final state.

Thank you very much for the careful discussion provided. The argument provided addresses my question in a satisfactory manner.

I have no further major concerns (other than the dubious significance of the experimental results). While sceptical of the interpretation offered for some observations, I think the authors have proven they are rigorous and I trust they will make sure to clarify questions regarding the structure of the preconditions and conditions of conditional effects in the benchmarks.

**Ethical Considerations:**

(1) Not Applicable: The paper does not have any ethical considerations to address